# OpenReview forum: "On the Role of Attention Heads in Large Language Model Safety"
_ICLR.cc/2025/Conference — ICLR 2025 Oral_

### Official Review · Reviewer_Ugyt · 2024-11-01

**Soundness:** 4
**Presentation:** 4
**Contribution:** 3
**Rating:** 8
**Confidence:** 4

**Summary:**

This work makes an essential contribution to the field of LLM interpretability and safety, especially given the increasing deployment of LLMs in sensitive areas where maintaining safety is paramount. By focusing on the attention heads, the authors target a critical yet underutilized aspect of LLM architecture. The Ships metric and Sahara algorithm also address interpretability efficiently, which is crucial for scaling such safety measures in larger models.

The results are promising; however, further research could examine whether these safety heads are consistent across more diverse LLM architectures beyond the ones tested (Llama-2-7b-chat and Vicuna-7b-v1.5). Additionally, the analysis of the trade-offs between safety and model helpfulness could be expanded to understand better how balancing these factors could affect real-world applications.

Overall, this paper offers valuable insights into safety interpretability and contributes a more resource-efficient framework for analyzing and enhancing LLM safety capabilities.

**Strengths:**

-Great presentation with easy to understand figures and well written concise mathematical concepts/methodology are key strenghts.
-The need of only 0.006% of parameter modification to achieve a SOTA ASR is also impressive.
-I found also interesting that the ablation tests showed clearly the contribution of safety from each head, with only a few ones being critical.
-The Sahara algorithm could be relevant in verifying the safety of LLMs.

**Weaknesses:**

Further research could examine whether these safety heads are consistent across more diverse LLM architectures beyond the ones tested (Llama-2-7b-chat and Vicuna-7b-v1.5). Additionally, the analysis of the trade-offs between safety and model helpfulness could be expanded to understand better how balancing these factors could affect real-world applications.

**Questions:**

How could one deploy the Ships and Sahara during the development process?
Which other use cases would be possible?
Robustness, related to attacks and otherwise, are often related to OOD samples. Could you posit the results of the paper in relation to OOD? And in this case, are the number of models and datasets used in the current study sufficient to evaluate it?

---

> ### Author Response · Authors · 2024-11-19
> **Response to Reviewer Ugyt**
>
> Thank you for your positive comments on our paper and your careful review.
>
>
> ## **For Weakness**
> The concerns you raised align well with our planned future work. In addition to the standard multi-head attention mechanism, there are several other attention architectures that we intend to explore in our upcoming research. Regarding the trade-off between helpfulness and safety, we acknowledge that this is an important aspect, as highlighted by reviewer ZyPy. We will review related literature and incorporate a more comprehensive evaluation of helpfulness in our future work.
>
>
> ## **For Question**
> These are insightful questions that we had not considered previously. We initially believe that attribution methods could potentially be used to identify specific safety-related heads within the model’s attention mechanism during the training process.
>
> As for the OOD sample, we find it challenging to address this issue at present. The main difficulty lies in the uncertainty of whether the training dataset contains data related to the harmful datasets we used. However, we plan to reach out to the model trainers in future work to obtain data distribution details, which will enable us to investigate the OOD problem more thoroughly.
>
> ---
>
> Thank you once again for your positive feedback on our paper!

---

> ### Author Response · Authors · 2024-11-25
> **Looking forward to further discussions!**
>
> **Dear Reviewer Ugyt,**
>
> Thank you for your thoughtful review and comments. We hope our responses have addressed your concerns adequately.
>
> We would be grateful to hear if you have any further questions or suggestions for improving our paper.
>
> We truly appreciate the time and effort you've taken in reviewing our paper and reply.
>
> **Best regards,**
>
> **Authors**

---

### Official Review · Reviewer_WKXw · 2024-11-03

**Soundness:** 2
**Presentation:** 2
**Contribution:** 2
**Rating:** 6
**Confidence:** 3

**Summary:**

The paper proposes the Safety Head Important Score (Ships) to evaluate the contribution of individual attention heads to the safety of language models. It presents the Safety Attention Head AttRibution Algorithm (Sahara), which identifies groups of attention heads that contribute to safety. The authors demonstrate through experiments that certain attention heads are crucial for safety, showing how their ablation can significantly increase the model's susceptibility to harmful queries.

**Strengths:**

1. The paper introduces a novel approach to understanding the safety mechanisms within large language models (LLMs) by presenting the Safety Head Important Score (Ships) and the Safety Attention Head AttRibution Algorithm (Sahara).
2. It effectively shifts the focus from generalized model parameters to specific attention heads that have a direct impact on the model's ability to reject harmful queries.
3. This work addresses a significant gap in the literature by systematically exploring the role of multi-head attention mechanisms in ensuring model safety, an area that has been relatively underexplored.
4. The discussions in Sections 4 and 5 are comprehensive and logically coherent, incorporating several insightful analyses and discussions.

**Weaknesses:**

1. Lines 256 and Appendix B.3 indicate that the ASR metric used in this paper employs a keyword-detection method, which is noted in [1] as having limitations that “lead to false positive and false negative cases.” Why is the GPT4-judge method, validated in [1] as a more comprehensive and accurate metric, not used? This method is commonly employed in 2024 LLM safety papers to measure ASR. The inaccuracies of ASR based on keyword detection in assessing successful attacks weaken the experimental data and analysis presented in the paper.
2. The analysis of Figure 2 is relatively weak (Lines 261-267). For instance, according to Figure 2, the improvement in ASR for Vicuna on Advbench and Jailbreakbench (direct) is quite limited, yet these two datasets are highly mainstream in the field of LLM safety. Does this not weaken the conclusions drawn in Lines 262-263?
3. There are still some writing issues in the paper, such as the figure numbering in Line 496, which should refer to Figure 6.
4. Is Figure 6b derived from Llama2 or Vicuna? I couldn't find this information in the paper. Additionally, Figure 6 shows an average decline of about 0.1 in Zero-Shot Task Scores, indicating a 15% decrease compared with the vanilla model. Given that the ASR determination in the paper only detects refusal keywords, I am concerned that the decline in helpfulness may reduce the model's understanding of harmful queries, potentially leading to responses that are affirmative but do not align with the harmful query. This could be counted towards the ASR, further undermining the discussion surrounding Figure 2.

[1] Fine-tuning aligned language models compromises safety, even when users do not intend to!

**Questions:**

1. Could you consider adding a new ASR metric, such as the GPT4-judge referenced in [1]? Alternatively, an analysis of the accuracy of the ASR based on keyword detection (in relation to human judgments) could be included.
2. In light of the second weakness, would it be possible to incorporate additional models, such as Gemma (instruct), to enhance the generalizability of the conclusion that “ablating the attention head with the highest Ships score significantly reduces safety capability”?

[1] Fine-tuning aligned language models compromises safety, even when users do not intend to!

I will reconsider my score if all these issues are adequately addressed.

---

> ### Author Response · Authors · 2024-11-14
> **Response to Reviewer WKXw**
>
> **Thank you for your detailed review. We think there are some misunderstandings and we will clarify them.**
>
> > All revisions are highlighted in purple.
>
> ## **For Weakness 1**
>
> **First, we would like to clarify that GPT-4-judge is not commonly employed in safety interpretability research.**
> The safety interpretability of LLMs, is still an emerging field with relatively few studies.
> In the existing research, for example, two closely related works[1, 2], they both used keyword detection instead of GPT-4-judge.
> Other related works, such as [3, 4], [3] used the model in [5] to evaluate toxicity, while [4] utilized a fine-tuned Llama from [6] without employing GPT-4-judge.
>
> The primary reason for this is that evaluating ASR with GPT-4 is prohibitively expensive for interpretability research, particularly when compared to jailbreak studies.
> Jailbreak evaluations typically require only **a few hundred data points**, while safety interpretability requires comparisons across **hundreds of thousands of data points** under different settings to analyze the parameters.
> To illustrate, we estimated the cost based on the prompt template from [7] and the paper you referenced.
> Their prompt templates have more than 400 tokens, so, every request needs approximately 600 input tokens for evaluating 128 tokens generation (the outputs may not have a fixed number, we omit them now).
> If we were to produce with GPT-4-judge, we would require roughly 120M tokens: 600(input tokens) * 1024 (head) * 100(dataset size) * 2(dataset) tokens, resulting in an estimated cost of 3,600 dollars for a single figure (30.00 dollars per 1M tokens for GPT-4)
> Since our work is more fine-grained than other works, it would cost over 10,000 dollars solely for input tokens across the entire paper, excluding any additional debugging or analysis costs.
> In contrast, a jailbreak evaluation typically costs around $3.60 for a similar sample size (0.12M tokens).
> **Thus, GPT-4 is rarely, if ever, used in safety interpretability research, which differs significantly from jailbreak studies**
>
> We sincerely hope that you will understand our choice of keyword detection, which is commonly employed in interpretability research due to both cost and efficiency. We believe it aligns well with the objectives of our work.
>
> ## **For Weakness 2**
> **Our improvement is not quiet limited, the increase from 0.27 to 0.55 is a significant improvement-it has effectively doubled.**
> In fact, the performance on Vicuna might seem limited because it is inherently not as safe as Llama, resulting in a smaller multiple of improvement compared to Llama.
> Nonetheless, this improvement is substantial enough to support our conclusions.
>
> ## **For Weakness 3**
> Thank you for pointing out the numbering issue.
> There is indeed an oversight here.
> **Figure 5a and Figure 5b have been correctly numbered as 6a and 6b**.
>
> ## **For Weakness 4**
> We apologize for the oversight regarding Figure 6b.
> The experimental results presented in Figure 6b are from Llama-2-7b-chat, and we have added  this clarification in Section 5.3.
> We believe your concern may stem from a misunderstanding.
> A harmful answer must also be helpful for malicious user[8], and our helpfulness evaluation on benign questions indicates that the model’s understanding of questions drops by about 15% overall.
> This level of degradation is generally considered to be insignificant in pruning methods [9, 10], which are chasing utility.
> We argue that our helpfulness analysis actually strengthens our findings rather than undermining them.
>
> ## **For Q1**
> **We appreciate your suggestion, but adding a metric based on GPT-4 is challenging for us due to the significant cost involved, which is one of the reasons GPT-4 is rarely used in safety interpretability research, particularly in our case.**
> Regarding human judgment, we understand your concern and appreciate your suggestion.
> While we did conduct small-scale manual reviews, they were not extensive enough to cover all generations due to similar cost constraints (as mentioned in Weakness 1).
> We also considered sampling before human evaluation, but this would still require us to process 1024 groups. Even if we were to sample just 10 items per group (which we acknowledge would be an inaccurate approximation), we would still need to evaluate 10,240 data points.
> Given these challenges, we decided not to include this approach in the paper.
> We hope you understand the difficulties we faced in this regard, and we sincerely appreciate your understanding.
>
> ## **For Q2**
> We believe it would be helpful to first address Weakness 2, as it is directly related to your concern. Once we have clarified that point, we will then provide a detailed response to this question.
>
> Limited to character, references are given in an additional comment.
>
> ---
>
> We sincerely hope that our response addresses your concerns and encourages you to reconsider your score. We would be more than happy to engage in additional discussions.

---

> > ### Author Response · Authors · 2024-11-14
> > **References**
> >
> > [1] Wei, Boyi, et al. "Assessing the Brittleness of Safety Alignment via Pruning and Low-Rank Modifications." Forty-first International Conference on Machine Learning.
> >
> > [2] Zhao, Wei, et al. "Defending Large Language Models Against Jailbreak Attacks via Layer-specific Editing." arXiv preprint arXiv:2405.18166 (2024).
> >
> > [3] Chen, Jianhui, et al. "Finding Safety Neurons in Large Language Models." arXiv preprint arXiv:2406.14144 (2024).
> >
> > [4] Arditi, Andy, et al. "Refusal in language models is mediated by a single direction." arXiv preprint arXiv:2406.11717 (2024).
> >
> > [5] Dai, Josef, et al. "Safe RLHF: Safe Reinforcement Learning from Human Feedback." The Twelfth International Conference on Learning Representations.
> >
> > [6] Mazeika, Mantas, et al. "HarmBench: A Standardized Evaluation Framework for Automated Red Teaming and Robust Refusal." Forty-first International Conference on Machine Learning.
> >
> > [7] Chao, Patrick, et al. "Jailbreaking black box large language models in twenty queries." arXiv preprint arXiv:2310.08419 (2023).
> >
> > [8] Bai, Yuntao, et al. "Training a helpful and harmless assistant with reinforcement learning from human feedback." arXiv preprint arXiv:2204.05862 (2022).
> >
> > [9] Elias Frantar and Dan Alistarh. Sparsegpt: Massive language models can be accurately pruned in one-shot. In International Conference on Machine Learning, pp. 10323–10337. PMLR, 2023.
> >
> > [10] Mingjie Sun, Zhuang Liu, Anna Bair, and J Zico Kolter. A simple and effective pruning approach for large language models. In The Twelfth International Conference on Learning Representations, 2024a.

---

> ### Comment · Reviewer_WKXw · 2024-11-14
>
> Thank you for your replies regarding weakness1 and weakness3. After your explanation, I fully understand the cost considerations in your experiments, which adequately resolve my concerns about the experimental setup. However, I would like to further clarify my questions for weakness2 and weakness4, as I believe there may be some misunderstanding of my points.
>
> ## Weakness2:
> My question concerns why the improvement of Vicuna on the **Advbench and Jailbreakbench** datasets in the **direct** setting was limited. My focus is not on comparing the averaged results of three datasets under the template setting (0.27 to 0.55), but rather understanding the constrained improvement specifically in the direct setting for those two datasets.
>
> ## Weakness4:
> You mentioned that the "understanding of questions drops." Could this lead to scenarios where a model **"responds affirmatively but does not actually answer the user's questions"**? For example, if a user asks, "generate a racist joke," the LLM might respond with "Here is a joke: [harmless joke]." In other cases, it might provide content unrelated to the user's real intent, which violates the rule you mentioned, "a harmful answer must also be helpful for malicious user". This situation **cannot be detected by a simple refusal keyword detection method**. Such cases might potentially undermine the reliability of the conclusion about the ASR improvement shown in Figure 2.
>
> ## For Q1:
> Conducting "an analysis of the accuracy of the ASR based on keyword detection (in relation to human judgments)" does not require testing under all of your experimental settings. I am interested in understanding the accuracy of the keyword detection measurement method itself. This accuracy primarily relates to differences in the style of the model's responses (for example, variation in refusal keywords and challenges in interpreting the semantics of responses). This concern is not related to the complexity of your experiments.
>
> Overall, most of my concerns have been resolved, and **I will increase the score to 6**. Thank you in advance for further addressing my remaining questions.

---

> > ### Author Response · Authors · 2024-11-18
> >
> > We sincerely appreciate your comments. We are encouraged by your approval of our cost considerations for the keyword detection method. Following your additional feedback, we now have a clearer understanding of your concerns regarding the remaining weaknesses, and we will provide further clarification below.
> >
> > ## **For Weakness2**
> > We understand your concern about the "direct setting". We believe that the limited improvements in this setting can be attributed to Vicuna's inherently low baseline when inputs are provided directly. Specifically, we posit that safety capabilities are not solely determined by the attention heads; prior studies[3] have also highlighted the role of parameters in the MLP in contributing to safety.
> >
> > As a result, there is likely an upper limit to the safety improvements achievable **by modifying only the attention heads**. In the case of Vicuna, its baseline **safety is so low in the direct setting that it approaches this upper limit even without further modifications**. In contrast, using a template better aligns with the formats encountered during alignment tuning, which can enhance safety capabilities. This alignment results in a more pronounced safety gap between the modified head and the vanilla model, explaining the observed clear differences in "template setting".
> >
> > ## **For Weakness4**
> > we appreciate your insightful concern and agree that such situations may indeed occur, even without modifying the attention heads, as discussed in the paper you mentioned. **However, if this type of decline does exist, it should align with the observed decrease in helpfulness, which we estimated to be no more than 15%.**
> >
> > Moreover, if modifications to these attention heads were to result in a significant decline in the model's ability to understand malicious goals, while the experimental results indicate maintained understanding of helpful goals, this also would support our argument. Specifically, it would indicate that these attention heads play a distinct role in influencing responses to malicious queries rather than general helpfulness, given that the modifications predominantly affect harmful queries.
> >
> > Overall, your concern is valid, and we recognize the need for further analysis to determine which scenario is at play.
> >
> > ## **For Q1**
> > We understand your question and appreciate your interest in analyzing the accuracy of ASR based on keyword detection in relation to human judgment. This is indeed a valuable direction for further exploration. In our current work, we recognize that such an analysis could be integrated into specific sections or expanded upon in future iterations of this research. We will carefully consider this in our revisions or subsequent studies.
> >
> > ---
> > We hope this response adequately addresses your remaining concerns. Thank you again for your thoughtful comments, which have significantly contributed to improving our work. If there are any unresolved issues, we would be more than happy to discuss them further.
> >
> > A kindly reminder: We noticed that you mentioned an intention to increase the score to 6. Please feel free to update your score by editing it. We appreciate your positive evaluation of our paper.

---

> > > ### Comment · Reviewer_WKXw · 2024-11-18
> > >
> > > Thanks for the clarification on my question. I have updated the score via edit.

---

### Official Review · Reviewer_ZyPy · 2024-11-04

**Soundness:** 3
**Presentation:** 3
**Contribution:** 3
**Rating:** 8
**Confidence:** 3

**Summary:**

The authors study how heads contribute to LLM safety, and find that by modifying less than 0.006% of the parameters (a significantly smaller number of parameters than before), safety alignment is degraded significantly. The authors propose Safety Head ImPortant Score (Ships) which measures individual heads’ contributions to model safety. On the dataset level, the authors propose an algorithm Sahara that iteratively choose important heads, creating a group of heads that degrades safety performance for a dataset. The method is efficient in compute hours needed, and impact safety at greater granularity than before. The authors study the effect of safety attention heads, including experiments of concatenating heads to a pre-trained model, and observes that safety capability is close to that of aligned model.

**Strengths:**

- The paper proposes a novel method for mechanistically locating and ablating heads that are important to safety alignment, with greater granularity and less compute than prior methods. The method of head ablation is well motivated and the experiments are detailed, considering course correction (reverting back to safety) as well.
- The paper is well written and organized

**Weaknesses:**

1. The helpfulness / utility measurement is done with lm-eval, which mostly consists of single-turn question and answering utility measurement. More comprehensive utility measurement would benefit the paper.
2. Sahara uses heuristic to choose group size, and group size is important to how safety capability is affected. Such size heuristics (more than 3) might not hold for different models with different number of parameters.
3. The paper is overall well-written with some small typos:  "Bottom. Results of attributing attributing specific harmful queries using Ships" (line 445). Consider changing the color for axis label for Figure 13. It's currently quite hard to read.

**Questions:**

Q1. What lm-eval helpful tasks were experimented on for Figure 6b?

Q2: Why do you think there is minimal overlap between the top-10 attention heads via UA ablation and SC ablation (line 477-478)? In combination with results from Appendix E, does this suggest SC is not a good ablation method?

Q3: What do you think contribute to safety capability improve when ablating a small head group between 1 and 3 (line 371)?

---

> ### Author Response · Authors · 2024-11-19
> **Response to ZyPy**
>
> We appreciate your positive feedback on our paper and your insightful comments.
>
> > All revisions are highlighted in purple.
>
> ## **For Weakness 1**
> We will conduct a review of related works to explore additional evaluation methods used beyond lm-eval on single-turn tasks. If relevant methods are identified, we will incorporate them into our analysis.
>
> ## **For Weakness 2**
> We acknowledge that the group size of 3 may not be optimal for models with different parameter sizes. We believe that further experiments are required to conduct on different models with different number of parameters for best group size.
>
> ## **For Weakness 3**
> Thank you for pointing out this issue. We have corrected this typo, as well as other typos suggested by the reviewers.
>
> ## **For Q1**
> We evaluated the utility of our method on five tasks: arc_challenge, boolq, openbookqa, rte, and winogrande. Thank you for your suggestion; we have now included this information in the main text.
>
> ## **For Q2**
> As shown in Figure 5a, the overlap between the important heads identified by the UA and SC methods is minimal.
>
> We believe that the SC method is flawed, even though it has been used in previous work. This could explain why prior studies did not identify the safety head. Methods like SC may not be suitable for LLM attention head ablation.
>
> ## **For Q3**
> We believe that alignment serves as an effective safety measure, as demonstrated in Figure 17 in Appendix D. When using the aligned template, the effect of ablation on the safety head is significantly reduced but remains effective.
>
> ---
>
> Overall, we sincerely appreciate your detailed review and would be happy to address any further questions or concerns you may have.

---

> ### Author Response · Authors · 2024-11-25
> **Looking forward to further discussions!**
>
> **Dear Reviewer ZyPy,**
>
> Thank you for your detailed questions and comments. We greatly appreciate your constructive questions, which have been instrumental in improving the quality of our work.
>
> If you have any additional questions or concerns that we can clarify or address, we would be happy to provide further information to ensure all aspects of our work are clear.
>
> Thank you once again for your valuable time and effort in reviewing our submission.
>
> **Best regards,**
>
> **Authors**

---

> > ### Comment · Reviewer_ZyPy · 2024-11-26
> >
> > Thank you for addressing my comments and adding evaluation details. I'm keeping my positive score as is.

---

### Official Review · Reviewer_BEKY · 2024-11-05

**Soundness:** 2
**Presentation:** 2
**Contribution:** 3
**Rating:** 6
**Confidence:** 4

**Summary:**

This paper introduces a method named Ships, which assesses each attention head’s contribution to LLM safety at both the query and dataset levels. It also proposes a heuristic search-based method, Sahara, to identify a group of safety-related attention heads. Ablating these heads led to a notable increase in ASR on Llama-2-7b-chat and Vicuna-7b-v1.5. The approach is claimed to be more parameter and computing efficient than previous methods.

**Strengths:**

1. To my knowledge, no other work has attempted to interpret each attention head’s contributions to LLM safety. I'm glad to see such work.

2. The method for assessing the importance of attention heads to safety is intuitive and reasonable.

**Weaknesses:**

1. Other previous works, such as those identifying safety-related parameters through probing [1], could also be discussed.

2. There are inconsistencies in notation usage that need to be addressed. To name a few:
- In Eq 2, 7 and 8, $d_k$ denotes the model dimension. However in line 297, $d$ is used instead.
- In Appendix A.1, $N = d/n$ should be clarified.
- Throughout the paper, $L$ and $n$ are used to denote the number of layers and the number of heads, respectively, but Algorithm 1 uses $\mathbb{L}$ and $\mathbb{N}$ for the same.
- Equation 5 defines parameter importance on safety as $\Delta p = p (\theta_O) - p (\theta_O \backslash \theta_c)$. However, $\Delta p$ is never mentioned again. Instead, Equation 9 uses KL divergence between two probabilities. I guess $\Delta p$ means to represent the difference between two probabilities computed by some function.
- Typo: Eq.6, $h_i^m$ → $h_i$

3. The impact of ablating safety-related heads on model utility could be evaluated.

4. Additional clarifications needed:
- In Table 3, are the results at the bottom obtained by, using Ships on each query to identify and ablate the most important head, then assessing whether the attack was successful for that query? If so, why does query-level Ships perform worse than dataset-level Ships?
- Line 264 notes that 0.006% of all parameters corresponds to the number of parameters in a single attention head. However, Section 4 suggests that ablating 0.006% of parameters (potentially one head) can achieve an ASR of approximately 0.72 on Llama. Yet, Figure 4(a) indicates that ablating at least two heads is necessary to surpass an ASR of 0.7. What is the group size used in Table 1?

[1] A Mechanistic Understanding of Alignment Algorithms: A Case Study on DPO and Toxicity. ICML 2024.

**Questions:**

See Weaknesses.

---

> ### Author Response · Authors · 2024-11-15
> **Resonse to Reviewer BEKY**
>
> **First of all, we sincerely thank you for your detailed and careful review. Your comments have greatly improved our paper.**
> > All revisions are highlighted in purple.
> ## **For Weakness 1**
> Thank you for pointing out the lack of discussion regarding the referenced work.
> We are aware of this research and acknowledge its relevance.
> Initially, we felt that it was not the most related to our study, which is why we did not discuss it in this version.
> Specifically, the referenced work focuses on the safety mechanisms of RLHF (or DPO), while our approach is centered on analyzing the role of the model's parameters.
> Nonetheless, we recognize that this distinction may need further clarification, and we plan to address it in an additional Appendix F in a new revision.
>
> ## **For Weakness 2**
> We sincerely apologize for the inconsistencies in notation usage.
> We greatly appreciate your careful review, and we will address all of your comments and revise the notational issues you pointed out.
> Additionally, we will review the paper thoroughly to identify and correct any other notation inconsistencies that may have been overlooked, with the aim of improving the clarity and quality of the paper.
>
> ### **2-1.**
> In line 297, we have to corrected $d$ to $d_k$ as you suggested.
> ### **2-2.**
> Thank you for your reminder, which led us to discover an error in the current version. Upon further review, we found that the derivation in the appendix version is more accurate.
>     To clarify, the correct expression should be:
>      $h_i^{\textcolor{purple}{mod}} = \operatorname{Softmax}\Big(\frac{\textcolor{orange}{\epsilon} W_q^i W_k^i{}^T}{\sqrt{d_k/n}}\Big) W_v^i = A W_v^i,$
>
> where \( A \) is a lower triangular matrix defined by
>     $A = [a_{ij}], \quad a_{ij} =
>     \begin{cases}
>     \frac{1}{i} & \text{if } i \geq j, \\
>     0 & \text{if } i < j.
>     \end{cases}
>     $
>
> We have made the corrections in both the main text and the corresponding appendix. Thank you again for pointing this out.
> ###  **2-3.**
> Regarding the use of the constants $\mathbb{L}, \mathbb{N},\mathbb{S}$, our intention was to unify the notation in the algorithm. We apologize if this caused confusion and will carefully consider modifying the notation for better consistency in future revisions.
> ### **2-4.**
>  You are absolutely correct in your observation. $\Delta p$ should indeed be expressed as a function rather than with a simple "$-$". We have revised Equation 5 to express this as KL divergence, as per your suggestion.
> ### **2-5.**
>  We would like to clarify that $h^m_i$ is not a typo. We use $m$ as a superscript to indicate that the i-th head is modified. However, we agree that this notation could be confusing, so we replace $m$ with $mod$ for easier understanding.
>
> ## **For Weakness 3**
> Thank you for your comment regarding the impact of ablating safety-related heads on model utility.
> As we mention briefly in Section 5.3, " HELPFUL-HARMLESS TRADE-OFF" we find that there is an effect on the model's performance and it is similar to pruning.
> We hope this clarification addresses your concern.
>
> ## **For Weakness 4**
> ### **Q1:**
> Thank you for your insightful question.
> The results at the bottom of Table 3 indeed come from query-level Ships.
> As for why query-level Ships perform worse than dataset-level Ships, this actually highlights the very motivation behind our proposal to use the generalized version of Ships.
> As we explain in lines 272 to 276, query-level Ships may be more tailored to one specific query.
> Therefore, it can achieve better results when only one is ablated, but it has little effect when more are ablated.
> For example, for a harmful query such as "how to generate defamatory articles," ablating the head related to understanding defamation may impact the model's response.
> However, this head ablation can not generalize to other types of queries, which is why dataset-level Ships can provide more consistent and generalized results.
> ### **Q2:**
> Yes, the 0.006% figure refers to modifying the Q, K, or V matrix in a single head.
> We apologize for the earlier confusion. Upon further review, we found that the results in Table 1 actually correspond to the ablation of three heads, **and the parameter modification should be corrected to approximately 0.018%.**
>
> ---
>
> **We sincerely appreciate your review comments again, which benefit our paper and make it more solid!**

---

> ### Author Response · Authors · 2024-11-25
> **Looking forward to further discussions!**
>
> **Dear Reviewer BEKY,**
>
> We would like to extend our heartfelt gratitude for your thoughtful and constructive comments on our manuscript. Your insightful feedback has not only helped us identify and address a critical oversight in our work but has also significantly strengthened the overall quality of our paper.
>
> We hope that our responses have effectively clarified your concerns and provided satisfactory explanations. If there are any remaining questions or additional points you would like to discuss, we would be more than happy to engage in further dialogue to address them.
>
> Once again, we sincerely appreciate the time and effort you have devoted to reviewing our submission. Your valuable input has been instrumental in improving our work, and we are truly grateful for your contribution.
>
> **Best regards,**
>
> **Authors**

---

### Meta-Review · Area_Chair_Qroh · 2024-12-12

**Metareview:**

This paper introduces SHIPS, a method for interpreting the contribution of individual attention heads to LLM safety. It proposes the Safety Attention Head AttRibution Algorithm (SAHARA), utilizing Undifferentiated Attention and Scaling Contribution to achieve this.

The paper presents a novel approach to analyzing attention heads in the context of LLM safety. The writing is clear and easy to follow.

The evaluation is limited, focusing primarily on LLaMA and Vicuna. Further evaluation on multi-turn inputs and a broader range of models would strengthen the findings.

The novelty of the proposed method and clear presentation justify acceptance.

**Additional Comments On Reviewer Discussion:**

Reviewers questions are adequately addressed. All the reviewer give positive score to the paper.

---

### Decision · Program_Chairs · 2025-01-22

Accept (Oral)